# An adaptive nearest neighbor rule for classification

**Akshay Balsubramani**
abalsubr@stanford.edu

**Sanjoy Dasgupta**
dasgupta@eng.ucsd.edu

**Yoav Freund**
yfreund@eng.ucsd.edu

**Shay Moran**
shaym@princeton.edu

## Abstract

We introduce a variant of the $k$-nearest neighbor classifier in which $k$ is chosen adaptively for each query, rather than being supplied as a parameter. The choice of $k$ depends on properties of each neighborhood, and therefore may significantly vary between different points. For example, the algorithm will use larger $k$ for predicting the labels of points in noisy regions.

We provide theory and experiments that demonstrate that the algorithm performs comparably to, and sometimes better than, $k$-NN with an optimal choice of $k$. In particular, we bound the convergence rate of our classifier in terms of a local quantity we call the "advantage", giving results that are both more general and more accurate than the smoothness-based bounds of earlier nearest neighbor work. Our analysis uses a variant of the uniform convergence theorem of Vapnik-Chervonenkis that is for empirical estimates of conditional probabilities and may be of independent interest.

## 1 Introduction

We introduce an adaptive nearest neighbor classification rule. Given a training set with labels $\{\pm 1\}$, its prediction at a query point $x$ is based on the training points closest to $x$, rather like the $k$-nearest neighbor rule. However, the value of $k$ that it uses can vary from query to query. Specifically, if there are $n$ training points, then for any query $x$, the smallest $k$ is sought for which the $k$ points closest to $x$ have labels whose average is either greater than $+\Delta(n, k)$, in which case the prediction is $+1$, or less than $-\Delta(n, k)$, in which case the prediction is $-1$; and if no such $k$ exists, then "?" ("don't know") is returned. Here, $\Delta(n, k) \sim \sqrt{(\log n)/k}$ corresponds to a confidence interval for the average label in the region around the query.

We study this rule in the standard statistical framework in which all data are i.i.d. draws from some unknown underlying distribution $P$ on $\mathcal{X} \times \mathcal{Y}$, where $\mathcal{X}$ is the data space and $\mathcal{Y}$ is the label space. We take $\mathcal{X}$ to be a separable metric space, with distance function $d : \mathcal{X} \times \mathcal{X} \to \mathbb{R}$, and we take $\mathcal{Y} = \{\pm 1\}$. We can decompose $P$ into the marginal distribution $\mu$ on $\mathcal{X}$ and the conditional expectation of the label at each point $x$: if $(X, Y)$ represents a random draw from $P$, define $\eta(x) = \mathbb{E}(Y|X = x)$. In this terminology, the Bayes-optimal classifier is the rule $g^* : \mathcal{X} \to \{\pm 1\}$ given by

$$g^*(x) = \begin{cases} \text{sign}(\eta(x)) & \text{if } \eta(x) \neq 0 \\ \text{either } -1 \text{ or } +1 & \text{if } \eta(x) = 0 \end{cases} \tag{1}$$

and its error rate is the Bayes risk, $R^* = \frac{1}{2}\mathbb{E}_{X \sim \mu}\left[1 - |\eta(X)|\right]$. A variety of nonparametric classification schemes are known to have error rates that converge asymptotically to $R^*$. These include $k$-nearest neighbor (henceforth, $k$-NN) rules [FH51] in which $k$ grows with the number of training points $n$ according to a suitable schedule $(k_n)$, under certain technical conditions on the metric measure space $(\mathcal{X}, d, \mu)$.

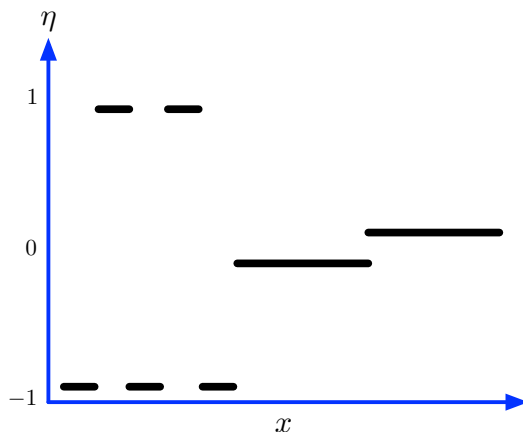

Figure 1: For values of $x$ on the left half of the shown interval, the pointwise bias $\eta(x)$ is close to $-1$ or 1, and thus a small value of $k$ will yield an accurate prediction. Larger $k$ will not do as well, because they may run into neighboring regions with different labels. For values of $x$ on the right half of the interval, $\eta(x)$ is close to 0, and thus large $k$ is essential for accurate prediction.

In this paper, we are interested in consistency as well as rates of convergence. In particular, we find that the adaptive nearest neighbor rule is also asymptotically consistent (under the same technical conditions) while converging at a rate that is about as good as, and sometimes significantly better than, that of $k$-NN under any schedule $(k_n)$.

Intuitively, one of the advantages of $k$-NN over nonparametric classifiers that use a fixed bandwidth or radius, such as Parzen window or kernel density estimators, is that $k$-NN automatically adapts to variation in the marginal distribution $\mu$: in regions with large $\mu$, the $k$ nearest neighbors lie close to the query point, while in regions with small $\mu$, the $k$ nearest neighbors can be further afield. The adaptive NN rule that we propose goes further: it also adapts to variation in $\eta$. In certain regions of the input space, where $\eta$ is close to 0, an accurate prediction would need large $k$. In other regions, where $\eta$ is near $-1$ or 1, a small $k$ would suffice, and in fact, a larger $k$ might be detrimental because neighboring regions might be labeled differently. See Figure 1 for one such example. A $k$-NN classifier is forced to pick a single value of $k$ that trades off between these two contingencies. Our adaptive NN rule, however, can pick the right $k$ in each neighborhood separately.

Our estimator allows us to give rates of convergence that are tighter and more transparent than those customarily obtained in nonparametric statistics. Specifically, for any point $x$ in the instance space $\mathcal{X}$, we define a notion of the *advantage at* $x$, denoted $\mathrm{adv}(x)$, which is rather like a local margin. We show that the prediction at $x$ is very likely to be correct once the number of training points exceeds $\tilde{O}(1/\mathrm{adv}(x))$. Universal consistency follows by establishing that almost all points have positive advantage.

## 1.1 Relation to other work in nonparametric estimation

For linear separators and many other *parametric* families of classifiers, it is possible to give rates of convergence that hold without any assumptions on the input distribution $\mu$ or the conditional expectation function $\eta$. This is not true of nonparametric estimation: although any target function can in principle be captured, the number of samples needed to achieve a specific level of accuracy will inevitably depend upon aspects of this function such as how fast it changes [DGL96, chapter 7]. As a result, nonparametric statistical theory has focused on (1) asymptotic consistency, ideally without assumptions, and (2) rates of convergence under a variety of smoothness assumptions.

Asymptotic consistency has been studied in great detail for the $k$-NN classifier, when $k$ is allowed to grow with the number of data points $n$. The risk of the classifier, denoted $R_n$, is its error rate on the underlying distribution $P$; this is a random variable that depends upon the set of training points seen. Cover and Hart [CH67] showed that in general metric spaces, under the assumption that every $x$ in the support of $\mu$ is either a continuity point of $\eta$ or has $\mu(\{x\}) > 0$, the expected risk $\mathbb{E}R_n$ converges to the Bayes-optimal risk $R^*$, as long as $k \to \infty$ and $k/n \to 0$. For points

in finite-dimensional Euclidean space, a series of results starting with Stone [Sto77] established consistency without any assumptions on $\mu$ or $\eta$, and showed that $R_n \to R^*$ almost surely [DGKL94]. More recent work has extended these *universal consistency* results—that is, consistency without assumptions on $\eta$—to arbitrary metric measure spaces $(\mathcal{X}, d, \mu)$ that satisfy a certain differentiation condition [CG06, CD14].

Rates of convergence have been obtained for $k$-nearest neighbor classification under various smoothness conditions including Holder conditions on $\eta$ [KP95, Gyö81] and "Tsybakov margin" conditions [MT99, AT07, CD14]. Such assumptions have become customary in nonparametric statistics, but they leave a lot to be desired. First, they are uncheckable: it is not possible to empirically determine the smoothness given samples. Second, they view the underlying distribution $P$ through the tiny window of two or three parameters, obscuring almost all the remaining structure of the distribution that also influences the rate of convergence. Finally, because nonparametric estimation is often *local*, there is the intriguing possibility of getting different rates of convergence in different regions of the input space: a possibility that is immediately defeated by reducing the entire space to two smoothness constants.

The first two of these issues are partially addressed by the work of [CD14], who analyze the finite sample risk of $k$-NN classification without any assumptions on $P$. Their bounds involve terms that measure the probability mass of the input space in a carefully defined region around the decision boundary: that is, bounds that are tailored to the specific distribution $P$, rather than reflecting worst-case behavior over some large class to which $P$ belongs. However, the expressions for the risk are somewhat hard to parse, in large part because of the interaction between $n$ and $k$.

In the present paper, we obtain finite-sample rates of convergence that are fine-tuned not just to the specific distribution $P$ but also to the specific query point. This is achieved by defining a *margin*, or *advantage*, at every point in the input space, and giving bounds (Theorem 1) entirely in terms of this quantity. For parametric classification, it has become common to define a notion of margin that controls generalization. In the nonparametric setting, it makes sense that the margin would in fact be a function $\mathcal{X} \to \mathbb{R}$, and would yield different generalization error bounds in different regions of space. Our adaptive nearest neighbor classifier allows us to realize this vision in a fairly elementary manner.

The advantages of setting $k$ locally have been pointed out and quantified in recent work on nonparametric *regression* [DGKL94, CS18], notably that of [Kpo11]. Although it is common to reduce classification to regression in nonparametric analysis, the right choice of $k$ may be fundamentally different in the two settings. This is reflected in the difference between our setting for $k$ and that of [Kpo11]; for instance, the physical value of the radius containing $k$ points matters in that work while playing no role in ours. Moreover, the benefit of local adaptivity may be more pronounced for classification than for regression. Our analysis shows, for instance, that there is a radius $r_x$ around each point $x$ such that prediction based on training points in $B(x, r_x)$ will with high probability be perfect, provided there are enough such points. This is not true of regression, where the target $y$ is a real value and thus the radius needs to keep shrinking.

**Organization.**  Most proofs are relegated to the appendices.

In Section 2, we introduce the formal model of learning and define some basic geometric notions, as a prelude to presenting the adaptive $k$-NN algorithm in Section 3. In Sections 4 and 5 and Appendix A, we state and prove consistency and generalization bounds for this classifier, and compare them with prior work in the $k$-NN literature. Our bounds exploit a general VC-based uniform convergence statement which is presented in Section 6 and proved in a self-contained manner in Appendix B.

## 2  Setup

Take the instance space to be a separable metric space $(\mathcal{X}, d)$ and the label space to be $\mathcal{Y} = \{\pm 1\}$. All data are assumed to be drawn i.i.d. from a fixed unknown distribution $P$ over $\mathcal{X} \times \mathcal{Y}$.

Let $\mu$ denote the marginal distribution on $\mathcal{X}$: if $(X, Y)$ is a random draw from $P$, then

$$\mu(S) = \Pr(X \in S)$$

for any measurable set $S \subseteq \mathcal{X}$. For any $x \in \mathcal{X}$, the conditional expectation, or *bias*, of $Y$ given $x$, is

$$\eta(x) = \mathbb{E}(Y|X = x) \in [-1, 1].$$

Similarly, for any measurable set $S$ with $\mu(S) > 0$, the conditional expectation of $Y$ given $X \in S$ is

$$\eta(S) = \mathbb{E}(Y|X \in S) = \frac{1}{\mu(S)} \int_S \eta(x)\, d\mu(x).$$

The risk of a classifier $g : \mathcal{X} \to \{-1, +1, ?\}$ is the probability that it is incorrect on pairs $(X, Y) \sim P$,

$$R(g) = P(\{(x, y) : g(x) \neq y\}). \tag{2}$$

The Bayes-optimal classifier $g^*$, as given in (1), depends only on $\eta$, but its risk $R^*$ depends on $\mu$. For a classifier $g_n$ based on $n$ training points from $P$, we will be interested in whether $R(g_n)$ converges to $R^*$, and the rate at which this convergence occurs.

The algorithm and analysis in this paper depend heavily on the probability masses and biases of balls in $\mathcal{X}$. For $x \in \mathcal{X}$ and $r \geq 0$, let $B(x, r)$ denote the closed ball of radius $r$ centered at $x$,

$$B(x, r) = \{z \in \mathcal{X} : d(x, z) \leq r\}.$$

For $0 \leq p \leq 1$, let $r_p(x)$ be the smallest radius $r$ such that $B(x, r)$ has probability mass at least $p$, that is,

$$r_p(x) = \inf\{r \geq 0 : \mu(B(x, r)) \geq p\}. \tag{3}$$

It follows that $\mu(B(x, r_p(x))) \geq p$.

The *support* of the marginal distribution $\mu$ plays an important role in convergence proofs and is formally defined as

$$\text{supp}(\mu) = \{x \in \mathcal{X} : \mu(B(x, r)) > 0 \text{ for all } r > 0\}.$$

It is a well-known consequence of the separability of $\mathcal{X}$ that $\mu(\text{supp}(\mu)) = 1$ [CH67].

## 3 The adaptive $k$-nearest neighbor algorithm

The algorithm is given a labeled training set $(x_1, y_1), \ldots, (x_n, y_n) \in \mathcal{X} \times \mathcal{Y}$. Based on these points, it is able to compute empirical estimates of the probabilities and biases of different balls.

For any set $S \subseteq \mathcal{X}$, we define its empirical count and probability mass as

$$\#_n(S) = |\{i : x_i \in S\}|$$
$$\mu_n(S) = \frac{\#_n(S)}{n}. \tag{4}$$

If this is non-zero, we take the empirical bias to be

$$\eta_n(S) = \frac{\sum_{i:x_i \in S} y_i}{\#_n(S)}. \tag{5}$$

The adaptive $k$-NN algorithm (AKNN) is shown in Figure 2. It makes a prediction at $x$ by growing a ball around $x$ until the ball has significant bias, and then choosing the corresponding label. In some cases, a ball of sufficient bias may never be obtained, in which event "?" is returned. In what follows, let $g_n : \mathcal{X} \to \{-1, +1, ?\}$ denote the AKNN classifier.

Later, we will also discuss a variant of this algorithm in which a modified confidence interval,

$$\Delta(n, k, \delta) = c_1 \sqrt{\frac{d_0 \log n + \log(1/\delta)}{k}} \tag{7}$$

is used, where $d_0$ is the VC dimension of the family of balls in $(\mathcal{X}, d)$.

In comparing the algorithm of Figure 2 to standard $k$-nearest neighbor classification, it might at first glance seem that we have merely replaced one parameter ($k$) with another ($\delta$). This is not accurate. Our $\delta$ is the customary confidence parameter of statistics and learning theory: it provides an upper bound on the failure probability of the algorithm. It can be set to $0.05$, for instance. The algorithm makes infinitely many parameter choices—it sets $k$ for each query point—and asks for just a single failure probability that lets it know how aggressively to set its confidence intervals.

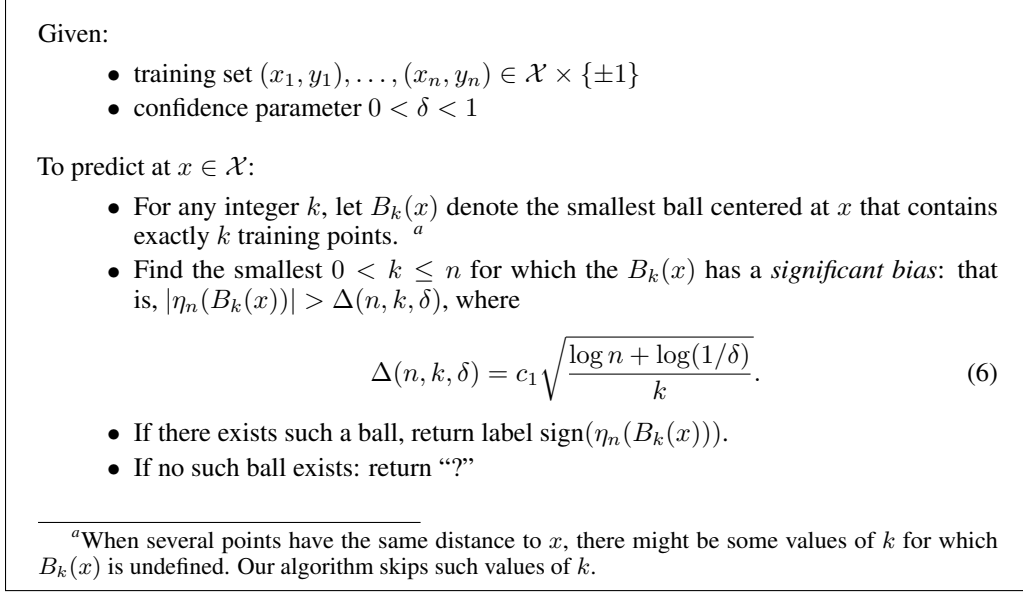

Given:

- training set $(x_1, y_1), \dots, (x_n, y_n) \in \mathcal{X} \times \{\pm 1\}$
- confidence parameter $0 < \delta < 1$

To predict at $x \in \mathcal{X}$:

- For any integer $k$, let $B_k(x)$ denote the smallest ball centered at $x$ that contains exactly $k$ training points. [a]
- Find the smallest $0 < k \le n$ for which the $B_k(x)$ has a *significant bias*: that is, $|\eta_n(B_k(x))| > \Delta(n, k, \delta)$, where

$$\Delta(n, k, \delta) = c_1 \sqrt{\frac{\log n + \log(1/\delta)}{k}}. \qquad (6)$$

- If there exists such a ball, return label $\text{sign}(\eta_n(B_k(x)))$.
- If no such ball exists: return "?"

---

[a]When several points have the same distance to $x$, there might be some values of $k$ for which $B_k(x)$ is undefined. Our algorithm skips such values of $k$.

Figure 2: The adaptive $k$-NN (AKNN) classifier. The absolute constant $c_1$ is from Lemma 7.

## 4 Pointwise advantage and rates of convergence

We now provide finite-sample rates of convergence for the adaptive nearest neighbor rule. For simplicity, we give convergence rates that are specific to any query point $x$ and that depend on a suitable notion of the "margin" of distribution $P$ around $x$.

Pick any $p, \gamma > 0$. Recalling definition (3), we say a point $x \in \mathcal{X}$ is $(p, \gamma)$-salient if the following holds for either $s = +1$ or $s = -1$:

- $s\eta(x) > 0$, and $s\eta(B(x, r)) > 0$ for all $r \in [0, r_p(x))$, and $s\eta(B(x, r_p(x))) \ge \gamma$.

In words, this means that $g^*(x) = s$ (recall that $g^*$ is the Bayes classifier), that the biases of all balls of radius $\le r_p(x)$ around $x$ have the same sign as $s$, and that the bias of the ball of radius $r_p(x)$ has absolute value at least $\gamma$. A point $x$ can satisfy this definition for a variety of pairs $(p, \gamma)$. The *advantage* of $x$ is taken to be the largest value of $p\gamma^2$ over all such pairs:

$$\text{adv}(x) = \begin{cases} \sup\{p\gamma^2 : x \text{ is } (p, \gamma)\text{-salient}\} & \text{if } \eta(x) \neq 0 \\ 0 & \text{if } \eta(x) = 0 \end{cases} \qquad (8)$$

We will see (Lemma 3) that under a mild condition on the underlying metric measure space, almost all $x$ with $\eta(x) \neq 0$ have a positive advantage.

### 4.1 Advantage-based finite-sample bounds

We now state two generalization bounds for the adaptive nearest neighbor classifier. The first holds pointwise—it bounds the probability of error at a specific point $x$—while the second is the type of uniform convergence bound that is more standard in learning theory.

The following theorem shows that for every point $x$, if the sample size $n$ satisfies $n \gtrsim 1/\text{adv}(x)$, then the label of $x$ is likely to be $g^*(x)$, where $g^*$ is the Bayes optimal classifier. This provides pointwise convergence of $g(x)$ to $g^*(x)$ at a rate that is sensitive to the local geometry of $x$.

**Theorem 1** (Pointwise convergence rate). *There is an absolute constant $C > 0$ for which the following holds. Let $0 < \delta < 1$ denote the confidence parameter in the* AKNN *algorithm (Figure 2), and suppose the algorithm is used to define a classifier $g_n$ based on $n$ training points chosen i.i.d. from $P$. Then, for every point $x \in \text{supp}(\mu)$, if*

$$n \ge \frac{C}{\text{adv}(x)} \max\left(\log \frac{1}{\text{adv}(x)}, \log \frac{1}{\delta}\right)$$

*then with probability at least $1 - \delta$ we have that $g_n(x) = g^*(x)$.*

If we further assume that the family of all balls in the space has finite VC dimension $d_0$ then we can strengthen the guarantee to hold with high probability *simultaneously* for all $x \in \text{supp}(\mu)$. This is achieved by a modified version of the algorithm that uses confidence interval (7) instead of (6).

**Theorem 2** (Uniform convergence rate). *Suppose that the set of balls in $(\mathcal{X}, d)$ has finite VC dimension $d_0$, and that the algorithm of Figure 2 uses confidence interval (7) instead of (6). Then, with probability at least $1 - \delta$, the resulting classifier $g_n$ satisfies the following: for every point $x \in \text{supp}(\mu)$, if*

$$n \geq \frac{C}{\text{adv}(x)} \max \left( \log \frac{1}{\text{adv}(x)}, \ \log \frac{1}{\delta} \right)$$

*then $g_n(x) = g^*(x)$.*

A key step towards proving Theorems 1 and 2 is to identify the subset of $\mathcal{X}$ that is likely to be correctly classified for a given number of training points $n$. This follows the rough outline of [CD14], which gave rates of convergence for $k$-nearest neighbor, but there are two notable differences. First, we will see that the likely-correct sets obtained in that earlier work (for $k$-NN) are, roughly, subsets of those we obtain for the new adaptive nearest neighbor procedure. Second, the proof for our setting is considerably more streamlined; for instance, there is no need to devise tie-breaking strategies for deciding the identities of the $k$ nearest neighbors.

## 4.2 A comparison with $k$-nearest neighbor

For $a \geq 0$, let $\mathcal{X}_a$ denote all points with advantage greater than $a$:

$$\mathcal{X}_a = \{x \in \text{supp}(\mu) : \text{adv}(x) > a\}. \tag{9}$$

In particular, $\mathcal{X}_0$ consists of all points with positive advantage.

By Theorem 1, points in $\mathcal{X}_a$ are likely to be correctly classified when the number of training points is $\widetilde{\Omega}(1/a)$, where the $\widetilde{\Omega}(\cdot)$ notation ignores logarithmic terms. In contrast, the work of [CD14] showed that with $n$ training points, the $k$-NN classifier is likely to correctly classify the following set of points:

$$\mathcal{X}'_{n,k} = \{x \in \text{supp}(\mu) : \eta(x) > 0, \eta(B(x,r)) \geq k^{-1/2} \text{ for all } 0 \leq r \leq r_{k/n}(x)\}$$
$$\cup \{x \in \text{supp}(\mu) : \eta(x) < 0, \eta(B(x,r)) \leq -k^{-1/2} \text{ for all } 0 \leq r \leq r_{k/n}(x)\}.$$

Such points are $(k/n, k^{-1/2})$-salient and thus have advantage at least $1/n$. In fact,

$$\bigcup_{1 \leq k \leq n} \mathcal{X}'_{n,k} \subseteq \mathcal{X}_{1/n}.$$

In this sense, the adaptive nearest neighbor procedure is able to perform *roughly* as well as all choices of $k$ simultaneously. This is not a precise statement because of logarithmic factors (the sample complexity in Theorem 1 is $(1/a) \log(1/a)$ rather than $1/a$), and the resulting gap can be seen in our experiments.

# 5 Universal consistency

In this section we study the convergence of $R(g_n)$ to the Bayes risk $R^*$ as the number of points $n$ grows. An estimator is described as *universally consistent* in a metric measure space $(\mathcal{X}, d, \mu)$ if it has this desired limiting behavior for all conditional expectation functions $\eta$.

Earlier work [CD14] established the universal consistency of $k$-nearest neighbor (for $k/n \to 0$ and $k/(\log n) \to \infty$) in any metric measure space that satisfies the Lebesgue differentiation condition: that is, for any bounded measurable $f : \mathcal{X} \to \mathbb{R}$ and for almost all ($\mu$-a.e.) $x \in \mathcal{X}$,

$$\lim_{r \downarrow 0} \frac{1}{\mu(B(x,r))} \int_{B(x,r)} f \, d\mu = f(x). \tag{10}$$

This is known to hold, for instance, in any finite-dimensional normed space or any doubling metric space [Hei01, Chapter 1].

We will now see that this same condition implies the universal consistency of the adaptive nearest neighbor rule. To begin with, it implies that almost every point has a positive advantage.

**Lemma 3.** *Suppose metric measure space $(\mathcal{X}, d, \mu)$ satisfies condition (10). Then, for any conditional expectation $\eta$, the set of points*

$$\{x \in \mathcal{X} : \eta(x) \neq 0, \ \mathrm{adv}(x) = 0\}$$

*has zero $\mu$-measure.*

*Proof.* Let $\mathcal{X}' \subseteq \mathcal{X}$ consist of all points $x \in \mathrm{supp}(\mu)$ for which condition (10) holds true with $f = \eta$, that is, $\lim_{r \downarrow 0} \eta(B(x, r)) = \eta(x)$. Since $\mu(\mathrm{supp}(\mu)) = 1$, it follows that $\mu(\mathcal{X}') = 1$.

Pick any $x \in \mathcal{X}'$ with $\eta(x) \neq 0$; without loss of generality, $\eta(x) > 0$. By (10), there exists $r_o > 0$ such that
$$\eta(B(x, r)) \geq \eta(x)/2 \text{ for all } 0 \leq r \leq r_o.$$
Thus $x$ is $(p, \gamma)$-salient for $p = \mu(B(x, r_o)) > 0$ and $\gamma = \eta(x)/2$, and has positive advantage. $\square$

Universal consistency follows as a consequence; the proof details are deferred to Appendix A.

**Theorem 4** (Universal consistency)**.** *Suppose the metric measure space $(\mathcal{X}, d, \mu)$ satisfies condition (10). Let $(\delta_n)$ be a sequence in $[0, 1]$ with (1) $\sum_n \delta_n < \infty$ and (2) $\lim_{n \to \infty} (\log(1/\delta_n))/n = 0$. Let the classifier $g_{n,\delta_n} : \mathcal{X} \to \{-1, +1, ?\}$ be the result of applying the $\mathrm{AKNN}$ procedure (Figure 2) with $n$ points chosen i.i.d. from $P$ and with confidence parameter $\delta_n$. Letting $R_n = R(g_{n,\delta_n})$ denote the risk of $g_{n,\delta_n}$, we have $R_n \to R^*$ almost surely.*

## 6 Uniform convergence of empirical conditional measures

A key piece of our analysis is a uniform convergence bound for empirical estimates of *conditional* probabilities. We now discuss this bound in an abstract setting; further details are in Appendix B.

Let $P$ be a distribution over some space $X$, and let $\mathcal{A}, \mathcal{B}$ be two collections of events. Let $x_1, \ldots, x_n$ be independent samples from $P$. We would like to use these to estimate $P(A|B)$ simultaneously for all $A \in \mathcal{A}, B \in \mathcal{B}$. It is natural to consider the empirical estimates:

$$P_n(A|B) = \frac{\sum_i 1_{[x_i \in A \cap B]}}{\sum_i 1_{[x_i \in B]}}.$$

We study the approximation error of these estimates. Note that the case where $\mathcal{B} = \{X\}$ (i.e., in which one estimates $P(A)$ using $P_n(A)$ simultaneously for all $A \in \mathcal{A}$) is handled by the classical VC theory. Let us assume that both $\mathcal{A}, \mathcal{B}$ have VC dimension upper-bounded by some $d_0$.

To demonstrate the kinds of statements we would like, consider the case where each of $\mathcal{A}, \mathcal{B}$ contains only one event: $\mathcal{A} = \{A\}$, and $\mathcal{B} = \{B\}$, and set $\#_n(B) = \sum_i 1_{[x_i \in B]}$. A Chernoff bound implies that conditioned on the event that $\#_n(B) > 0$, the following holds with probability at least $1 - \delta$:

$$|P(A|B) - P_n(A|B)| \leq \sqrt{\frac{2\log(1/\delta)}{\#_n(B)}}. \tag{11}$$

This bound depends on $\#_n(B)$ and is thus data-dependent. To derive it, use that conditioned on $x_i \in B$, event $x_i \in A$ has probability $P(A|B)$, so random variable "$\#_n(B) \cdot p_n(A|B)$" has a binomial distribution with parameters $\#_n(B)$ and $P(A|B)$.

We would want to prove a uniform version of (11), of the form: with probability at least $1 - \delta$,

$$(\forall A \in \mathcal{A})\,(\forall B \in \mathcal{B}) : |P(A|B) - P_n(A|B)| \leq O\left(\sqrt{\frac{d_0 \log(1/\delta)}{\#_n(B)}}\right).$$

But as we explain in the appendix, this is unfortunately false. Instead, we prove the following (slightly weaker) variant:

**Theorem 5** (UCECM). *Let $P$ be a probability distribution over $X$, and let $\mathcal{A}, \mathcal{B}$ be two families of measurable subsets of $X$ such that $\mathsf{VC}(\mathcal{A}), \mathsf{VC}(\mathcal{B}) \leq d_0$. Let $n \in \mathbb{N}$, and let $x_1 \ldots x_n$ be $n$ i.i.d samples from $P$. Then the following event occurs with probability at least $1 - \delta$:*

$$(\forall A \in \mathcal{A})\,(\forall B \in \mathcal{B}) : |P(A|B) - P_n(A|B)| \leq \sqrt{\frac{k_o}{\#_n(B)}},$$

*where $k_o = 1000\,(d_0 \log(8n) + \log(4/\delta))$, and $\#_n(B) = \sum_{i=1}^{n} 1[x_i \in B]$.*

## 7   Experiments

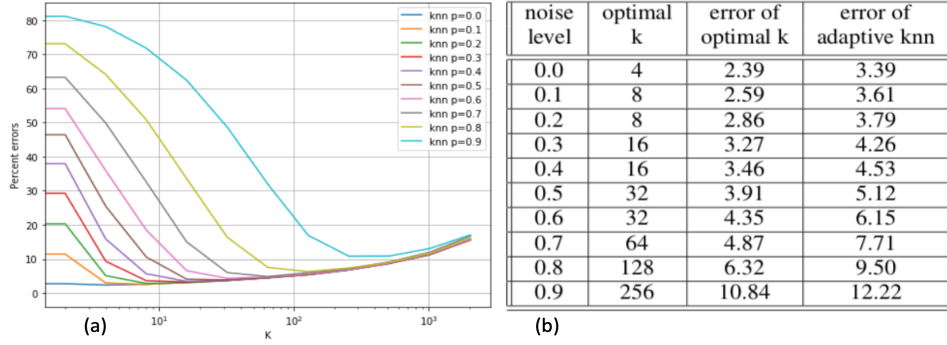

| noise level | optimal k | error of optimal k | error of adaptive knn |
|---|---|---|---|
| 0.0 | 4 | 2.39 | 3.39 |
| 0.1 | 8 | 2.59 | 3.61 |
| 0.2 | 8 | 2.86 | 3.79 |
| 0.3 | 16 | 3.27 | 4.26 |
| 0.4 | 16 | 3.46 | 4.53 |
| 0.5 | 32 | 3.91 | 5.12 |
| 0.6 | 32 | 4.35 | 6.15 |
| 0.7 | 64 | 4.87 | 7.71 |
| 0.8 | 128 | 6.32 | 9.50 |
| 0.9 | 256 | 10.84 | 12.22 |

Figure 3: Effect of label noise on $k$-NN and AKNN. Performance on MNIST for different levels of random label noise $p$ and for different values of $k$. Each line in the figure on the left **(a)** represents the performance of $k$-NN as a function of $k$ for a given level of noise. The optimal choice of $k$ increases with the noise level, and that the performance degrades severely for too-small $k$. The table **(b)** shows that AKNN, with a fixed value of $A$, performs almost as well as $k$-NN with the optimal choice of $k$.

We performed a few experiments using real-world data sets from computer vision and genomics (see Section C). These were conducted with some practical alterations to the algorithm of Fig. 2.

**Multiclass extension:** Suppose the set of possible labels is $\mathcal{Y}$. We replace the binary rule "find the smallest $k$ such that $|\eta_n(B_k(x))| > \Delta(n, k, \delta)$" with the rule: "find the smallest $k$ such that $\eta_n^y(B_k(x)) - \frac{1}{|\mathcal{Y}|} > \Delta(n, k, \delta)$ for some $y \in \mathcal{Y}$, where $\eta_n^y(S) \doteq \frac{\#_n\{x_i \in S \text{ and } y_i = y\}}{\#_n(S)}$."

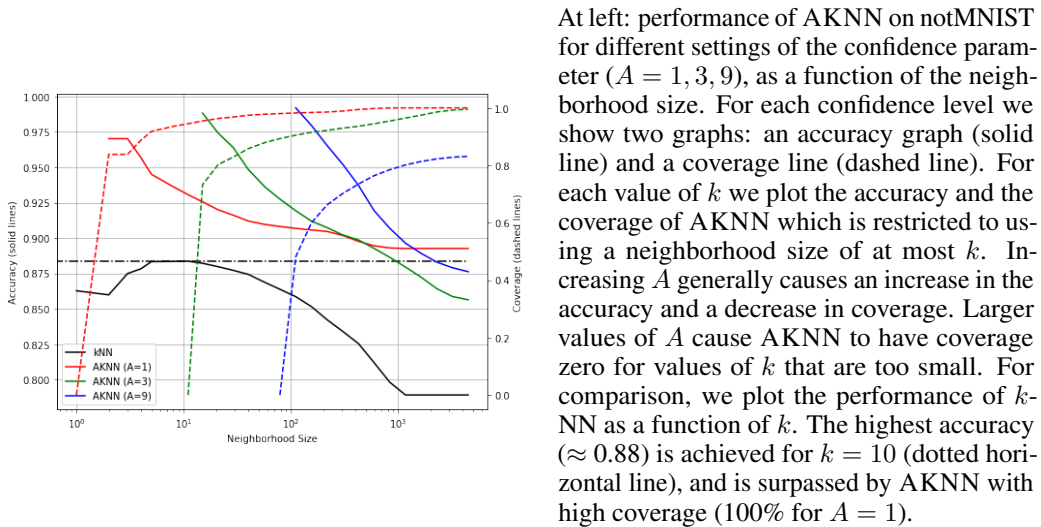

At left: performance of AKNN on notMNIST for different settings of the confidence parameter ($A = 1, 3, 9$), as a function of the neighborhood size. For each confidence level we show two graphs: an accuracy graph (solid line) and a coverage line (dashed line). For each value of $k$ we plot the accuracy and the coverage of AKNN which is restricted to using a neighborhood size of at most $k$. Increasing $A$ generally causes an increase in the accuracy and a decrease in coverage. Larger values of $A$ cause AKNN to have coverage zero for values of $k$ that are too small. For comparison, we plot the performance of $k$-NN as a function of $k$. The highest accuracy ($\approx 0.88$) is achieved for $k = 10$ (dotted horizontal line), and is surpassed by AKNN with high coverage (100% for $A = 1$).

Figure 4: Performance of AKNN on notMNIST. See also Figure 5.

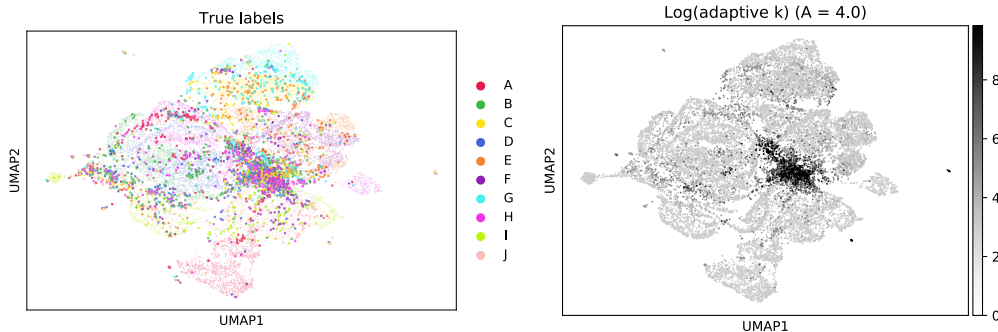

Figure 5: A visualization of the performance of AKNN on notMNIST. **(a)** The correct labels, with prediction errors of AKNN ($A = 4$) highlighted. **(b)** The value of $k$ chosen by the algorithm when predicting each datapoint.

**Parametrization:** We replace Equation (6) with $\Delta = \frac{A}{\sqrt{k}}$, where $A$ is a confidence parameter corresponding to the theory's $\delta$ (given $n$).

**Resolving multilabel predictions:** Our algorithm can output answers that are not a single label. The output can be "?", which indicates that no label has sufficient evidence. It can also be a subset of $\mathcal{Y}$ that contains more than one element, indicating that more than one label has significant evidence. In some situations, using subsets of the labels is more informative. However, when we want to compare head-to-head with $k$-NN, we need to output a single label. We use a heuristic to predict with a single label $y \in \mathcal{Y}$ on any $x$: the label for which $\max_k \eta_n^y(B_k(x))/\sqrt{k}$ is largest.

We briefly discuss our main conclusions from the experiments, with more details in Appendix C.

**AKNN is comparable to the best $k$-NN rule.** In Section 4.2 we prove that AKNN compares favorably to $k$-NN with any fixed $k$. We demonstrate this in practice in different situations. With simulated independent label noise on the MNIST dataset (Fig. 3), a small value of $k$ is optimal for noiseless data, but performs very poorly when the noise level is high. On the other hand, AKNN adapts to the local noise level automatically, as demonstrated without adding noise on the more challenging notMNIST and single-cell genomics data (Fig. 4, 5, 6).

**Varying the confidence parameter $A$ controls abstaining.** The parameter $A$ controls how conservative the algorithm is in deciding to abstain, instead of incurring error by predicting. $A \to 0$ represents the most aggressive setting, in which the algorithm never abstains, essentially predicting according to a 1-NN rule. Higher settings of $A$ cause the algorithm to abstain on some of these predicted points, for which there is no sufficiently small neighborhood with a sufficiently significant label bias (Fig. 7).

**Adaptively chosen neighborhood sizes reflect local confidence.** The number of neighbors chosen by AKNN is a local quantity that gives a practical pointwise measure of the confidence associated with label predictions. Small neighborhoods are chosen when one label is measured as significant nearly as soon as statistically possible; by definition of the AKNN stopping rule, this is not true where large neighborhoods are necessary. In our experiments, performance on points with significantly higher neighborhood sizes dropped monotonically, with the majority of the data set having performance significantly exceeding the best $k$-NN rule over a range of settings of $A$ (Fig. 4, 6; Appendix C).

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
