[Supplementary Material · adaptive-knn.pdf]

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

$$n \geq \frac{C}{\mathrm{adv}(x)} \max\left(\log \frac{1}{\mathrm{adv}(x)}, \log \frac{1}{\delta}\right)$$

*then with probability at least $1 - \delta$ we have that $g_n(x) = g^*(x)$.*

If we further assume that the family of all balls in the space has finite VC dimension $d_0$ then we can strengthen the guarantee to hold with high probability *simultaneously* for all $x \in \mathrm{supp}(\mu)$. This is achieved by a modified version of the algorithm that uses confidence interval (7) instead of (6).

**Theorem 2** (Uniform convergence rate). *Suppose that the set of balls in $(\mathcal{X}, d)$ has finite VC dimension $d_0$, and that the algorithm of Figure 2 uses confidence interval (7) instead of (6). Then, with probability at least $1 - \delta$, the resulting classifier $g_n$ satisfies the following: for every point $x \in \mathrm{supp}(\mu)$, if*

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

## Footnotes

[1]Indeed, Lemma 7 follows from Theorem 5 by plugging in $\mathcal{A} = \{\mathcal{X} \times \{+1\}\}, \mathcal{B} = \{C \times \{\pm 1\} : C \in \mathcal{C}\}$.

[2]This is motivated by finite-dimensional Euclidean space $\mathbb{R}^D$, where it holds with $d_0 = D + 1$ ([Dud79]).

[3]This follows from analyzing the maximal bin in a uniform assignment of $\Theta(n)$ balls into $n$ bins [RS98]

[4]Note that the above inequality makes sense also when $k(B) = 0$, by identifying $\frac{\cdot}{0}$ as $\infty$, and using the convention that $\infty - \infty = \infty$ and that $\infty \leq \infty$.

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

# A Analysis and proofs

The first step in establishing advantage-dependent rates of convergence is to bound the accuracy of empirical estimates of probability mass and bias. This is achieved by a careful choice of large deviation bounds.

## A.1 Large deviation bounds

Suppose we draw $n$ points $(x_1, y_1), \ldots, (x_n, y_n)$ from $P$. If $n$ is reasonably large, we would expect the empirical mass $\mu_n(S)$ of any set $S \subset \mathcal{X}$, as defined in (4), to be close to its probability mass under $\mu$. The following lemma, from [CD10], quantifies one particular aspect of this.

**Lemma 6** ([CD10], Lemma 7). *There is a universal constant $c_0$ such that the following holds. Let $\mathcal{B}$ be any class of measurable subsets of $\mathcal{X}$ of VC dimension $d_0$. Pick any $0 < \delta < 1$. Then with probability at least $1 - \delta^2/2$ over the choice of $(x_1, y_1), \ldots, (x_n, y_n)$, for all $B \in \mathcal{B}$ and for any integer $k$, we have*

$$\mu(B) \geq \frac{k}{n} + \frac{c_0}{n} \max\left(k, d_0 \log \frac{n}{\delta}\right) \implies \mu_n(B) \geq \frac{k}{n}.$$

Likewise, we would expect the empirical bias $\eta_n(S)$ of a set $S \subset \mathcal{X}$, as defined in (5), to be close to its true bias $\eta(S)$. The latter is defined whenever $\mu(S) > 0$.

**Lemma 7.** *There is a universal constant $c_1$ for which the following holds. Let $\mathcal{C}$ be a class of subsets of $\mathcal{X}$ with VC dimension $d_0$. Pick any $0 < \delta < 1$. Then with probability at least $1 - \delta^2/2$ over the choice of $(x_1, y_1), \ldots, (x_n, y_n)$, for all $C \in \mathcal{C}$,*

$$|\eta_n(C) - \eta(C)| \leq \Delta(n, \#_n(C), \delta)$$

*where $\#_n(C) = |\{i : x_i \in B\}|$ is the number of points in $C$ and*

$$\Delta(n, k, \delta) = c_1 \sqrt{\frac{d_0 \log n + \log(1/\delta)}{k}}. \tag{12}$$

Lemma 7 is a special case[1] of a uniform convergence bound for conditional probabilities (Theorem 5) that we prove in Section 6.

## A.2 Proof of Theorem 1

Theorem 1 is an immediate consequence of the following lemma, in which the choice of constants is made explicit.

**Lemma 8.** *Define $c_2 = \max(c_1, 1/2)\sqrt{1 + c_0}$, where $c_0$ and $c_1$ are the constants from Lemmas 6 and 7, and take $c_3 = 16c_2^2$. Pick any $x \in \mathrm{supp}(\mu)$ with $\mathrm{adv}(x) > 0$. Fix any $0 < \delta < 1$. If the number of training points satisfies*

$$n > \frac{c_3}{\mathrm{adv}(x)} \max\left(\log \frac{c_3}{\mathrm{adv}(x)}, \log \frac{1}{\delta}\right),$$

*then with probability at least $1 - \delta^2$ over the choice of training data, the adaptive nearest neighbor rule will have $g_n(x) = g^*(x)$.*

*Proof.* Pick any $x \in \mathrm{supp}(\mu)$. Suppose $\eta(x) > 0$; the negative case is symmetric. The set $\mathcal{B}$ of all balls centered at $x$ is easily seen to have VC dimension $d_0 = 1$. By Lemmas 6 and 7, we have that with probability at least $1 - \delta^2$, the following two properties hold for all $B \in \mathcal{B}$:

1. For any integer $k$, we have $\#_n(B) \geq k$ whenever $n\mu(B) \geq k + c_0 \max(k, \log(n/\delta))$.

2. $|\eta_n(B) - \eta(B)| \leq \Delta(n, \#_n(B), \delta)$.

Assume henceforth that these hold.

By the definition of advantage, point $x$ is $(p, \gamma)$-salient for some $p, \gamma > 0$ with $p\gamma^2 = \mathrm{adv}(x) - \epsilon$, where we can make $\epsilon > 0$ arbitrarily small. The lower bound on $n$ in the theorem statement then implies that

$$\gamma \geq 2c_2 \sqrt{\frac{\log n + \log(1/\delta)}{np}}, \tag{13}$$

or equivalently that $np\gamma^2 \geq 4c_2^2(\log n + \log(1/\delta))$.

Set $k = np/(1 + c_0)$. By (13) we have $np \geq 4c_2^2 \log(n/\delta)$ and thus $k \geq \log(n/\delta)$. As a result, $np \geq k + c_0 \max(k, \log(n/\delta))$, and by property 1, the ball $B = B(x, r_p(x))$ has $\#_n(B) \geq k$. This means, in turn, that by property 2,

$$\begin{aligned}
\eta_n(B) \geq \; &\eta(B) - \Delta(n, k, \delta) = \gamma - c_1 \sqrt{\frac{\log(n/\delta)}{k}} \\
&\geq 2c_2 \sqrt{\frac{\log(n/\delta)}{np}} - c_1 \sqrt{\frac{\log(n/\delta)}{k}} \geq 2c_1 \sqrt{\frac{\log(n/\delta)}{k}} - c_1 \sqrt{\frac{\log(n/\delta)}{k}} \\
&= c_1 \sqrt{\frac{\log(n/\delta)}{k}} \geq \Delta(n, \#_n(B), \delta).
\end{aligned}$$

Thus ball $B$ would trigger a prediction of $+1$.

At the same time, for any ball $B' = B(x, r)$ with $r < r_p(x)$,

$$\eta_n(B') \geq \eta(B') - \Delta(n, \#_n(B'), \delta) > -\Delta(n, \#_n(B'), \delta)$$

and thus no such ball will trigger a prediction of $-1$. Therefore, the prediction at $x$ must be $+1$. □

### A.3  Proof of Theorem 2

This proof follows much the same outline as that of Theorem 1. A crucial difference is that uniform large deviation bounds (Lemmas 6 and 7) are applied to the class of all balls in $\mathcal{X}$, which is assumed[2] to have finite VC dimension $d_0$. In contrast, the proof of Theorem 1 only applies these bounds to the class of balls centered at a specific point, which has VC dimension at most 1 in any metric space.

### A.4  Proof of Theorem 4

Recall from (9) that $\mathcal{X}_a$ denotes the set of points with advantage $> a$.

**Lemma 9.** *Let $c_3$ be the constant from Lemma 8. Pick any $0 < \delta < 1$ as a confidence parameter for the* AKNN *estimator of Figure 2. Fix any $a > 0$. If the number of training points $n$ satisfies*

$$n \geq \frac{c_3}{a} \max \left( \log \frac{c_3}{a}, \; \log \frac{1}{\delta} \right),$$

*then with probability at least $1 - \delta$, the resulting classifier $g_n$ has risk*

$$R(g_n) - R^* \leq \delta + \mu(\mathcal{X}_0 \setminus \mathcal{X}_a).$$

*Proof.* From Lemma 8, we have that for any $x \in \mathcal{X}_a$,

$$\mathrm{Pr}_n(g_n(x) \neq g^*(x)) \leq \delta^2,$$

where $\mathrm{Pr}_n$ denotes probability over the choice of training points. Thus, for $X \sim \mu$,

$$\mathbb{E}_n \mathbb{E}_X 1(g_n(X) \neq g^*(X)|X \in \mathcal{X}_a) \leq \delta^2,$$

and by Markov's inequality,

$$\mathrm{Pr}_n[\mathrm{Pr}_X(g_n(X) \neq g^*(X)|X \in \mathcal{X}_a) \geq \delta] \leq \delta.$$

Thus, with probability at least $1 - \delta$ over the training set,

$$\Pr_X(g_n(X) \neq g^*(X)|X \in \mathcal{X}_a) \leq \delta.$$

On points with $\eta(x) = 0$, both $g_n$ and the Bayes-optimal $g^*$ incur the same risk. Thus

$$
\begin{aligned}
R(g_n) - R^* &\leq \Pr_X(g_n(X) \neq g^*(X)|X \in \mathcal{X}_a) + \Pr_X(X \notin \mathcal{X}_a, \eta(X) \neq 0) \\
&\leq \delta + \Pr_X(X \in \mathcal{X}_0 \setminus \mathcal{X}_a) + \Pr_X(\mathrm{adv}(X) = 0, \eta(X) \neq 0) \\
&\leq \delta + \mu(\mathcal{X}_0 \setminus \mathcal{X}_a),
\end{aligned}
$$

where we invoke Lemma 3 for the last step. $\qquad\square$

We now complete the proof of Theorem 4. Given the sequence of confidence parameters $(\delta_n)$, define a sequence of advantage values $(a_n)$ by

$$a_n = \frac{c_3}{n} \max\left( 2 \log n, \ \log \frac{1}{\delta_n} \right).$$

The conditions on $(\delta_n)$ imply $a_n \to 0$.

Pick any $\epsilon > 0$. By the conditions on $(\delta_n)$, we can pick $N$ so that $\sum_{n \geq N} \delta_n \leq \epsilon$. Let $\omega$ denote a realization of an infinite training sequence $(X_1, Y_1), (X_2, Y_2), \ldots$ from $P$. By Lemma 9, for any positive integer $N$,

$$\Pr\left( \omega : \exists n \geq N \text{ s.t. } R(g_n(\omega)) - R^* > \delta_n + \mu(\mathcal{X}_0 \setminus \mathcal{X}_{a_n}) \right) \leq \sum_{n \geq N} \delta_n \leq \epsilon.$$

Thus, with probability at least $1 - \epsilon$ over the training sequence $\omega$, we have that for all $n \geq N$,

$$R(g_n(\omega)) - R^* \leq \delta_n + \mu(\mathcal{X}_0 \setminus \mathcal{X}_{a_n}),$$

whereupon $R(g_n(\omega)) \to R^*$ (since $\delta_n, a_n \to 0$ and $\lim_{a \downarrow 0} \mu(\mathcal{X}_0 \setminus \mathcal{X}_a) = 0$). Since this holds for any $\epsilon > 0$, the theorem follows.

## B Uniform Convergence of Empirical Conditional Measures

### B.1 Formal Statement

Let $P$ be a distribution over $X$, and let $\mathcal{A}, \mathcal{B}$ be two collections of events. Consider $n$ independent samples from $P$, denoted by $x_1, \ldots, x_n$. We would like to estimate $P(A|B)$ simultaneously for all $A \in \mathcal{A}, B \in \mathcal{B}$. It is natural to consider the empirical estimates:

$$P_n(A|B) = \frac{\sum_i 1_{[x_i \in A \cap B]}}{\sum_i 1_{[x_i \in B]}}.$$

We study when (and to what extent) these estimates provide a good approximation. Note that the case where $\mathcal{B} = \{X\}$ (i.e., in which one estimates $P(A)$ using $P_n(A)$ simultaneously for all $A \in \mathcal{A}$) is handled by the classical VC theory. Throughout this section we assume that both $\mathcal{A}, \mathcal{B}$ have finite VC dimension, and we let $d_0$ denote an upper bound on both $\mathsf{VC}(\mathcal{A})$ and $\mathsf{VC}(\mathcal{B})$.

To demonstrate the kinds of statements we would like to derive, consider the case where each of $\mathcal{A}, \mathcal{B}$ contains only one event: $\mathcal{A} = \{A\}$, and $\mathcal{B} = \{B\}$, and set $\#_n(B) = \sum_i 1_{[x_i \in B]}$. A Chernoff bound implies that conditioned on the event that $\#_n(B) > 0$, the following holds with probability at least $1 - \delta$:

$$|P(A|B) - P_n(A|B)| \leq \sqrt{\frac{2 \log(1/\delta)}{\#_n(B)}}. \tag{14}$$

To derive it, use that conditioned on $x_i \in B$, the event $x_i \in A$ has probability $P(A|B)$, and therefore the random variable "$\#_n(B) \cdot p_n(A|B)$" has a binomial distribution with parameters $\#_n(B)$ and $P(A|B)$.

Note that the bound on the error in Equation (14) depends on $\#_n(B)$ and therefore is data-dependent. We stress that this is the type of statement we want: the more samples belong to $B$, the more certain we are with the empirical estimate. Thus, we would want to prove a statement as follows:

With probability at least $1 - \delta$,

$$(\forall A \in \mathcal{A})\,(\forall B \in \mathcal{B}) : |P(A|B) - P_n(A|B)| \leq O\left(\sqrt{\frac{d_0 \log(1/\delta)}{\#_n(B)}}\right),$$

where $\#_n(B) = \sum_{i=1}^{n} 1[x_i \in B]$.

The above statement is, unfortunately, false. As an example, consider the probability space defined by drawing $x \sim [n]$ uniformly, and then coloring $x$ by $c_x \in \{\pm 1\}$ uniformly. For each $i$ let $B_i$ denote the event that $i$ was drawn, and let $A$ denote the event that the drawn color was $+1$. (formally, $B_i = \{i\} \times \{\pm 1\}$, and $A = [n] \times \{+1\}$). One can verify that the VC dimension of $\mathcal{B} = \{B_i : i \leq n\}$ and of $\mathcal{A} = \{A\}$ is at most 1. The above statement fails in this setting: indeed, one can verify that if we draw $n$ samples from this space then with a constant probability there will be some $j$ such that:

(i) $j$ always gets the same color (say $+1$), and

(ii) $j$ is sampled at least $\Omega(\log n / \log \log n)$ times[3].

Therefore, with constant probability we get that

$$P_n(A|B_i) = 1, P(A|B_i) = 1/2,$$

and so the difference between the error is clearly $1 - (1/2) = 1/2$, which is clearly not upper bounded by $O(\sqrt{\log \log n / \log n})$.

We prove the following (slightly weaker) variant:

**Theorem** (Theorem 5 restatement). *Let $P$ be a probability distribution over $X$, and let $\mathcal{A}, \mathcal{B}$ be two families of measurable subsets of $X$ such that $\mathsf{VC}(\mathcal{A}), \mathsf{VC}(\mathcal{B}) \leq d_0$. Let $n \in \mathbb{N}$, and let $x_1 \ldots x_n$ be $n$ i.i.d samples from $P$. Then the following event occurs with probability at least $1 - \delta$:*

$$(\forall A \in \mathcal{A})\,(\forall B \in \mathcal{B}) : |P(A|B) - P_n(A|B)| \leq \sqrt{\frac{k_o}{\#_n(B)}},$$

*where $k_o = 1000\,(d_0 \log(8n) + \log(4/\delta))$, and[4] $\#_n(B) = \sum_{i=1}^{n} 1[x_i \in B]$.*

**Discussion.** Theorem 5 can be combined with Lemma 6 to yield a bound on the minimal $n$ for which $P_n(A|B)$ is a non-trivial approximation of $P(A|B)$. Indeed, Lemma 6 implies that if $n$ is large enough so that $P(B) = \Omega\left(\frac{d_0 \log n}{n}\right)$, then the empirical estimate $P_n(A|B)$ is a decent approximation. In the context of the adaptive nearest neighbor classifier, this means that the empirical biases provide meaningful estimates of the true biases for balls whose measure is $\tilde{\Omega}\left(\frac{d_0}{n}\right)$. This resembles the learning rate in realizable settings.

We remark that a weaker statement than Theorem 5 can be derived as a corollary of the classical uniform convergence result [VC71]. Indeed, since the VC dimension of $\{B \cap A : i \in \mathcal{I}\}$ is at most $d_0$, it follows that

$$P_n(A|B) \approx \frac{P(A \cap B) \pm \sqrt{d_0/n}}{P(B) \pm \sqrt{d_0/n}}.$$

However, this bound guarantees non-trivial estimates only once $P(B)$ is roughly $\sqrt{d_0/n}$. This is similar to the learning rate in agnostic (i.e., non-realizable) settings.

Another major advantage of the uniform convergence bound in Theorem 5 is that it is data-dependent: if many points from the sample belong to $B \in \mathcal{B}$ (i.e. $\#_n(B)$ is large), then we get better guarantees on the approximation of $P(A|B)$ by $P_n(A|B)$ for all $A \in \mathcal{A}$.

## B.2 Proof of Theorem 5

As noted above, the standard uniform convergence bound for VC classes can not yield the bound in Theorem 5. Instead, we use a variant of it due to [BBL05] which concerns *relative deviations* (see [BBL05]: Theorem 5.1 and the discussion before Corollary 5.2). In order to state the theorem, we need the following notation: Let $\mathcal{C}$ be a family of subsets of $\mathcal{X}$. We denote by $\mathbb{S}_{\mathcal{C}} : \mathbb{N} \to \mathbb{N}$ the *growth function* of $\mathcal{C}$, which is defined by:

$$\mathbb{S}_{\mathcal{C}}(n) = \max\{|\mathcal{C}|_R| : R \subseteq X, |R| = n\},$$

where $\mathcal{C}|_R = \{C \cap R : C \in \mathcal{C}\}$ is the projection of $\mathcal{C}$ to $R$.

**Theorem 10** ([BBL05]). *Let $\mathcal{C}$ be a family of subsets of $X$ and let $P$ be a distribution over $\mathcal{X}$. Then, the following holds with probability $1 - \delta$:*

$$(\forall C \in \mathcal{C}) : |P(C) - P_n(C)| \leq 2\sqrt{P_n(C)\frac{\log \mathbb{S}_{\mathcal{C}}(2n) + \log(4/\delta)}{n}} + 4\frac{\log \mathbb{S}_{\mathcal{C}}(2n) + \log(4/\delta)}{n}.$$

Set $\mathcal{C} = \mathcal{B} \cup \{A \cap B : A \in \mathcal{A}, B \in \mathcal{B}\}$. We prove Theorem 5 by applying Theorem 10 on $\mathcal{C}$; to this end we first upper bound $\mathbb{S}_{\mathcal{C}}(n)$. Let $\mathcal{D} = \{A \cap B : A \in \mathcal{A}, B \in \mathcal{B}\}$, so that $\mathcal{C} = \mathcal{B} \cup \mathcal{D}$. Then:

$$\mathbb{S}_{\mathcal{C}}(n) \leq \mathbb{S}_{\mathcal{B}}(n) + \mathbb{S}_{\mathcal{D}}(n) \leq \mathbb{S}_{\mathcal{B}}(n) + \mathbb{S}_{\mathcal{A}}(n)\mathbb{S}_{\mathcal{B}}(n) \leq 2\mathbb{S}_{\mathcal{A}}(n)\mathbb{S}_{\mathcal{B}}(n) \leq 2\binom{n}{\leq d_0}^2 \leq 2(2n)^{2d_0},$$

where the second inequality follows since $\mathbb{S}_{\mathcal{D}}(n) \leq \mathbb{S}_{\mathcal{A}}(n)\mathbb{S}_{\mathcal{B}}(n)$, the second to last inequality follows from the Sauer-Shelah-Perles Lemma, and the last inequality follows since $\binom{a}{\leq b} \leq (2a)^b$. Therefore, applying Theorem 10 on $\mathcal{C}$ yields that with probability $1 - \delta$ the following event holds:

$$(\forall C \in \mathcal{C}) : |P(C) - P_n(C)| \leq 4\sqrt{P_n(C)\frac{d_0 \log 8n + \log(4/\delta)}{n}} + 8\frac{d_0 \log 8n + \log(4/\delta)}{n}. \quad (15)$$

For the remainder of the proof we assume that the event in Equation (15) holds and argue that it implies the conclusion in Theorem 5. Let $A \in \mathcal{A}$, $B \in \mathcal{B}$, and let $k = n \cdot P_n(B) = \#_n(B)$ denote the number of data points in $B$. We want to show that

$$|P(A|B) - P_n(A|B)| \leq \sqrt{\frac{k_o}{k}}, \quad (16)$$

where $k_o = 1000 \left(d_0 \log(8n) + \log(4/\delta)\right)$. Let $j = k \cdot P_n(A|B) = \#_n(A \cap B)$ denote the number of data points in $A \cap B$. We establish Equation (16) by showing that

$$P(A|B) \leq P_n(A|B) + \sqrt{\frac{k_o}{k}} \quad \text{and} \quad P(A|B) \geq P_n(A|B) - \sqrt{\frac{k_o}{k}}.$$

In the following calculation it will be convenient to denote $D := d_0 \log(8n) + \log(4/\delta)$. By Equation (15) we get:

$$P(A|B) = \frac{P(A \cap B)}{P(B)}$$

$$\leq \frac{P_n(A \cap B) + 4\sqrt{P_n(A \cap B)\frac{D}{n}} + 8\frac{D}{n}}{P_n(B) - 4\sqrt{P_n(B)\frac{D}{n}} - 8\frac{D}{n}}$$

$$= \frac{\frac{P_n(A\cap B)}{P_n(B)} + 4\sqrt{\frac{P_n(A\cap B)}{P_n(B)}\frac{D}{nP_n(B)}} + 8\frac{D}{nP_n(B)}}{1 - 4\sqrt{\frac{D}{nP_n(B)}} - 8\frac{D}{nP_n(B)}}s = P_n(A|B)\frac{1 + 4\sqrt{\frac{D}{j}} + 8\frac{D}{j}}{1 - 4\sqrt{\frac{D}{k}} - 8\frac{D}{k}},$$

where the first inequality follows from Equation (15) and the following equalities are trivial. Thus,

$$P(A|B) \leq \frac{j}{k}\left(\frac{1 + 4\sqrt{\frac{D}{j}} + 8\frac{D}{j}}{1 - 4\sqrt{\frac{D}{k}} - 8\frac{D}{k}}\right). \quad (17)$$

Next, note that we may assume that $k \geq k_o = 1000D$, as otherwise Equation (16) trivially holds. Therefore,

$$\frac{1}{1 - 4\sqrt{\frac{D}{k}} - 8\frac{D}{k}} \leq 1 + 8\sqrt{\frac{D}{k}} + 16\frac{D}{k}. \qquad \left((\forall x < \tfrac{1}{2}) : \frac{1}{1-x} \leq 1 + 2x\right)$$

Plugging this in Equation (17), and using first that $j \leq k$ and then that $1000D \leq k$, yields:

$$\begin{aligned}
P(A|B) &\leq \frac{j}{k}\left(1 + 4\sqrt{\frac{D}{j}} + 8\frac{D}{j}\right)\left(1 + 8\sqrt{\frac{D}{k}} + 16\frac{D}{k}\right) \\
&= \frac{j}{k} + 8\frac{j}{k}\sqrt{\frac{D}{k}}\left(1 + 2\sqrt{\frac{D}{k}}\right) + \left(\frac{4\sqrt{jD} + 8D}{k}\right)\left(1 + 4\sqrt{\frac{D}{k}}\right)^2 \\
&\leq \frac{j}{k} + 8\sqrt{\frac{D}{k}}\left(1 + 2\sqrt{\frac{D}{k}}\right) + \left(4\sqrt{\frac{D}{k}} + \frac{8D}{k}\right)\left(1 + 4\sqrt{\frac{D}{k}}\right)^2 \\
&\leq \frac{j}{k} + 30\sqrt{\frac{D}{k}} = P_n(A|B) + \sqrt{\frac{k_o}{k}},
\end{aligned}$$

and so

$$P(A|B) \leq P_n(A|B) + \sqrt{\frac{k_o}{k}}.$$

A symmetric argument yields similarly to Equation (17) that:

$$P(A|B) \geq \frac{j}{k}\left(\frac{1 - 4\sqrt{\frac{D}{j}} - 8\frac{D}{j}}{1 + 4\sqrt{\frac{D}{k}} + 8\frac{D}{k}}\right).$$

Then, a similar calculation (using the relation $(\forall x > 0) : \frac{1}{1+x} \geq 1 - 2x$) implies that

$$P(A|B) \geq P_n(A|B) - \sqrt{\frac{k_o}{k}},$$

which finishes the proof. $\qquad\square$

## C   Experimental Results

### C.1   Datasets

We test AKNN on several datasets. The first was the MNIST dataset of 70000 examples ([MNI96]).

We also evaluate AKNN on the more challenging notMNIST dataset ([not11]), consisting of extracted glyphs of the letters A-J from publicly available fonts. We use the 18724 labeled examples from this set, preprocessed feature-wise to be in $\left[-\frac{1}{2}, \frac{1}{2}\right]$ using $x \mapsto \frac{x}{255} - \frac{1}{2}$.

We further use AKNN on a challenging binary classification task of continuing interest, involving gene expression data on a population of single cells from different mouse organs collected by the Tabula Muris consortium ([C$^+$18], as processed in [Mou18]). This constitutes 45291 cells (training examples). Each cell has its data collected using one of two approaches. The task is to classify between them. More details follow.

The data are collected using representative protocols of two currently dominant approaches to isolate and measure single cells: a plate-based approach sorting cells into microwells, and a droplet-based approach manipulating cells using microfluidic technologies. Each approach has its own set of technical biases, about which much remains to be understood. Identifying and characterizing these biases to discriminate between such approaches is currently of great interest.

Both approaches measure effectively the same cells for our purposes, so there is a large decision boundary in the binary classification problem.

## C.2 A note on efficient implementation

In this paper, we computed the nearest neighbors of data exactly when running AKNN, to faithfully demonstrate its behavior. In practice, this would be done using approximate nearest-neighbor search to build a $k$-NN graph using a small fixed $k$ (say 10), and then using pairwise distances on this graph to compute neighborhoods as needed by AKNN. We tried this (using the nearest-neighbor method of [DCL11]) on notMNIST without substantive differences in the results.

## C.3 Supplemental Figures

Figure 6: As Fig. 4, on single-cell mouse data. AKNN is notably accurate on small-neighborhood points at moderate coverage, and performance drops off at higher $k$, with $A$ controlling this frontier.

Figure 7: AKNN predictions on notMNIST, for different settings of $A$.

Figure 8: AKNN neighborhood sizes on notMNIST, in increasing order of $A$, plotted on a log scale. Top left figure ($A = 0$) represents a 1-NN classifier. Bottom right figure ($A = 15$) shows that many of the points' neighborhoods are maximally large, which can be compared to the right panel of Fig. 7.

Figure 9: As Fig. 8, on single-cell mouse data, with the AKNN k-values replaced by their quantiles over the data. The relative ordering of the data by AKNN neighborhood size is fairly robust to $A$.