[Reviews · NeurIPS 2019]

Reviewer 1



After reading the author feedback, I am increasing my score as the authors have largely addressed my main concern of comparison/relating to existing locally adaptive NN estimators. The authors address this concern on the theory side. I still think additional numerical experiments to compare the proposed method with other adaptive NN approaches would strengthen the paper and be beneficial to practitioners (and could possibly lead to interesting theoretical questions if, for instance, there are some surprising performance gaps). Thanks to the authors for a well-thought-out response to the reviews! * * * * * Original review below * * * * * I found this paper very well-written and clear. The analysis is quite clean and elegant due to the newly introduced local notion of "advantage" as an alternative to the usual local Lipschitz conditions. The paper, however, does not discuss enough related work, especially adaptive nearest neighbor and kernel methods based on Lepski's method (for example, the adaptive k-NN method by Kpotufe (2011)). I realize that much of these existing methods are specified in the context of regression rather than classification but it's straightforward casting the regression results in terms of classification (by thresholding on the regression function, taken to be the conditional probability function). Mainly I would like to better understand fundamentally what is different about the strategy used by Kpotufe (2011) (as well as Lepski in many papers that are more focused on automatic bandwidth selection for kernel methods) vs your approach (I realize the theoretical analysis is quite different, and the proof ideas using advantage are much cleaner). Separately, there's also the $k^*$-NN approach for adaptively choosing k per data point (Anava and Levy 2016), for which theoretical analysis for a toy case is presented in Section 3.7 of Chen and Shah (2018). Is there some toy setup for which all three approaches could be analyzed to compare them theoretically? Separately, there are various works on adaptive k-NN classification for which the metric rather than k is adaptive (e.g., Short and Fukunaga 1981; Hastie and Tibshirani 1996; Domeniconi, Peng, and Gunopulos 2002; Wang, Neskovic, and Cooper 2007, Kpotufe, Boularias, Schultz, and Kim 2016). In experimental results, perhaps it would be helpful to benchmark against existing locally adaptive methods for choosing k (Kpotufe 2011; Anava and Levy 2016) as well as some simple sample splitting/cross validation kind of approach to selecting a single k, preferably on more datasets. In particular, it would be helpful to practitioners to understand how these locally adaptive nearest neighbor methods compare with one another, and when using a locally adaptive k yields a significant accuracy improvement over a global choice of k.

Reviewer 2



The algorithm is new. A novel concept was introduced to the theoretical analysis.

Reviewer 3



My main concern with this this paper is that, although the authors provide an algorithm that chooses k adaptively, they have introduced another tuning parameter \delta, and there is no guidance on how to choose this in practice. In the introduction the authors claim that their results are stanceoptimal.However,thereisnorest̲tˆprovestheoptimalityofthebounds.Therates of convergence' provided are not rates of convergence in the usual sense. They provide a threshold above which n must be for a given point x to be classified correctly with high probability. This says nothing about the average behaviour over test points x. The comparison with the standard k-NN classifier in Section 4.2 does not seem fully convincing to me. [CD14] show that the points in \mathcal{X}_{n,k}' are likely to be correctly classified by k-NN, but it is not clear whether there could be other points that are also likely to be correctly classified by k-NN. I think that more work needs to be done here to compare the methods convincingly. =========================================================== UPDATE (after author rebuttal): Thank you to the authors for carefully reading my review. I still feel that there should be more comparison to the literature on kNN classification, but my other concerns have now been mostly addressed. I have changed the score for the submission.

[Author Response · NeurIPS 2019]

# Author feedback: "An adaptive nearest neighbor rule for classification"

Our thanks to all the reviewers for their time and constructive comments.

**Reviewer 1**

We thank the reviewer for the careful and detailed comments. It is clear that the reviewer has understood our paper well.

In terms of the comparison to existing work: Thank you for pointing out some other papers that suggest methods for setting $k$ locally. We will add a discussion of these to our paper; the most relevant is probably Kpotufe (2011).

In a nutshell: (1) Even though classification is often reduced to regression in nonparametric analysis, methods of setting $k$ locally should be rather different in the two settings, and this is reflected in the large difference between Kpotufe's setting and ours (for instance, the physical value of the radius containing $k$ points actually matters in his setting but plays no role in ours); (2) what's more, the benefit of this local adaptivity is likely to be more pronounced in the case of classification. Our analysis, for instance, shows that in classification, there is a radius $r(x)$ around each point $x$ such that prediction based on $B(x, r)$ for any $r \leq r(x)$ will w.h.p. be perfect, provided enough points fall in this ball. (Once $n$ is large enough that there are enough points within this radius, you're done!) This is not, of course, true for regression, where the target $y$ is a real value and thus the radius needs to keep shrinking.

**Reviewer 2**

The reviewer asks for clarification on how the parameter $\delta$ (i.e. $A$, in practice) should be set. We will clarify this point further in the paper, and discuss theoretical considerations in the response to Reviewer 3. Briefly, we find in the paper that in practice, increasing $A$ lowers coverage and raises performance on the predicted set, but increases the neighborhood size required to predict rather than abstain. On the other hand, practical considerations typically imply that not too many neighbors can be used. So a practical $A$ should be as small as possible to achieve a desired coverage, with a given maximum neighborhood size per point.

**Reviewer 3**

We thank the reviewer for the detailed comments and address some of the questions raised.

1. *Setting the parameter $\delta$.* This is the standard confidence parameter of statistics and learning theory: it provides an upper bound on the failure probability of the algorithm. It can be set to 0.05, for instance. So it is not the case that we have replaced one parameter ($k$) with another ($\delta$). Rather, our algorithm automatically makes infinitely many parameter choices (we pick a different $k$ for each point) and asks for just a single failure probability that lets it know how aggressively to set its confidence intervals.

2. *Rates of convergence on single points vs the entire test distribution.* We provide both types of bounds: Theorem 1 gives a bound for individual points, whereas Theorem 2 provides the sort of uniform convergence bound that is more standard in learning theory.

3. *Instance optimality.* The reviewer is correct that claims of instance-optimality in the introduction are not substantiated later in the paper. This is indeed an omission, but it is not terribly complicated and we will add it to the paper.

In a nutshell: For any point $x$, the "advantage" $\mathrm{adv}(x)$ is equal to $p\gamma^2$, where there is a ball centered at $x$ that has probability mass $p$ and has average $y$-value that is either $\frac{1}{2} + \gamma$ or $\frac{1}{2} - \gamma$. Given only this information about $x$, in order to predict $x$'s label correctly with constant probability, we need $\gamma^{-2}$ points in the ball; thus we need $\Omega(1/(p\gamma^2)) = \Omega(1/\mathrm{adv}(x))$ data points overall.

[Meta-Review · NeurIPS 2019]

The paper proposes a variant of the k-Nearest Neighbors algorithm (called adaptive knn) in which $k$ is chosen for each example to classify, instead of being tuned as a global hyperparameter. To do so, the authors define a new notion that applied locally in the input space they call the advantage, instead of the local Lipschitz condition that is often used in such setting. An important contribution of the paper is the prove that the proposed algorithm is consistent and have pointwise convergence at the limit. The proposed notion of advantage is allers related to some error bounds for pointwise convergence. The experimental part is clearly sufficient for this type of paper, even if there is no comparison with other state-of-the-art algorithm. It nevertheless experimentally shows how performance increases when the proposed algorithm is compare with the classical knn. The reviewers and I consider that the theory developed in the paper is elegant and based on innovative ideas -- definitely not incremental. Also, the notion of advantage could be useful in other learning situations. For all these reason, I will recommend acceptation and a short talk. I think that it is in the interest of the community to ear about this idea of “advantage”. However, I highly encourage the authors to incorporate the ideas contained in their rebuttal to the camera ready since it clearly explained most of the concerns the reviewers had with the paper.